# Conjunction Ground Triangulation of Auroras and Magnetospheric Processes Observed by the Van Allen Probe Satellite near 6 Re

**Boris V. Kozelov [1,\*] and Elena E. Titova [1,2]**

[1] Polar Geophysical Institute, 184209 Apatity, Russia; lena.titova@gmail.com
[2] Space Research Institute of Russian Academy of Sciences, 84/32 Profsoyuznaya Street, 117997 Moscow, Russia
[\*] Correspondence: boris.kozelov@gmail.com

**Abstract:** Conjunction observations of auroras with electron distributions and broadband electrostatic fluctuations on Van Allen Probe A satellite in the equatorial region are considered. Using triangulation measurements, the energy spectra of the precipitating electrons in the rayed auroral structures were determined for the 17 March 2015 event. A comparison of the spectra of precipitating electrons in the auroral rays with satellite measurements of electrons in the equatorial region related to the aurora showed their agreement. The concomitance between Van Allen Probe A broadband electric waves and auroral variations measured by the ground-based auroral camera was observed on 17 March 2015. This suggests that broadband electrostatic waves may be responsible for electron precipitation, leading to the formation of rayed structures in the aurora.

**Keywords:** aurora; triangulation; energetic spectra of electrons; Van Allen Probe satellite; broadband electric waves





## 1. Introduction

It is well known that various wave modes interact with electrons at the equator and contribute to the scattering of electrons in the loss cone and the formation of auroras. Previous studies have identified several wave modes that can scatter equatorial electrons, including whistler-mode waves [1,2], electron cyclotron harmonic waves [3], and kinetic Alfvén waves [4].

These wave modes are predominantly observed from pre-midnight to dawn [5], directly associated with plasma sheet injections. The most intense electron injections are observed during substorms in the midnight sector. In recent years, observations by the Van Allen Probe (VAP) spacecraft have shown that these injections are often accompanied by intense broadband electrostatic waves in the inner equatorial magnetosphere [6]. The maximum amplitude of these waves is typically observed at frequencies below a few hundred hertz. The broadband electrostatic wave is often caused by nonlinear electrostatic spikes, including electron holes, double layers, and certain types of more complex solitary waves [7]. Recent theoretical and experimental studies have demonstrated that these broadband electrostatic waves can scatter low-energy ($\leq 1$ keV) electrons in the plasma layer [8], directly scatter electrons into the loss cone [9], generate enhanced streams of longitudinal electrons [10], and produce auroras [11]. During the development of a substorm, different wave modes are usually detected simultaneously on the VAP satellite. In such cases, it becomes challenging to determine the specific effects of electron scattering and precipitation associated with broadband electrostatic waves.

Fluxes of electrons precipitating from the magnetosphere cause an optical glow in the upper atmosphere—the aurora. Various forms of auroras are observed, many of which are well studied morphologically, such as arcs, pulsating spots, and diffuse auroras [12,13]. Detailed reports on the altitude of the auroral glow are rare in the literature. The method of such determination can be divided into two groups:

(1)   Determination of the glow height from simultaneous observations from different points [12–14];
(2)   Determination of the glow height from observations from one point in different auroral lines.

To implement the second approach, much more sophisticated equipment is needed to detect weak luminescence in different lines as well as significant theoretical assumptions about the physicochemical processes leading to the emission of these lines [15]. The first group includes triangulation and tomographic methods [12–14]. The solution to such a problem for the glow of a volume of gas in the general case is complicated in comparison with the solution for solid bodies in stereovision. In [16–18], a simple implementation of the stereo-triangulation approach has been used for observations in a relatively small field of view and with a small distance between cameras, which makes it possible to simplify the identification of structures in frames in case of the auroral ray observation. The altitude profile of the auroral emission rate and an estimate of the energy distribution in the auroral electron flux were obtained for several events of rayed structure observations.

In this paper, we will consider an event when weak auroral intensifications of rayed arcs were observed on the ground and only the broadband electrostatic waves were detected by VAP-A satellite in the conjugated region of the magnetosphere at ~6 Re.

## 2. Instruments

The data of the optical instruments of the Multiscale Auroral Imaging Network (MAIN) system for continuous monitoring of the night sky glow were used in the work: an all-sky camera and two narrow-angle cameras separated in the east-west direction by 4 km in Apatity, APT (67.57° N, 33.3° E) and the Apatity range APP (67.57° N, 33.4° E). Narrow-angle cameras are oriented towards the region of the local magnetic zenith in the sky; a small spatial separation makes it possible to obtain images of the glow from the heights of the range of 80–300 km at different angles, sufficient to determine the height of the glow structures in the atmosphere. More information about the equipment can be found in [19]. The calibration of narrow angle cameras using a test lamp is discussed in [20]. The use of high bandwidth glass filters in the blue-green region of visible light allowed us to use simple, cheap cameras to patrol record the aurora with time resolution of 1 s. Magnetic field measurements were carried out in Lovozero, LOZ (67.97° N, 35.02° E)

The Van Allen Probes have low-inclination (10°) orbit with 600 km perigee and 30,000 km apogee [21]. In this study, we used VLF wave and electron distribution measurements from Van Allen Probe A (VAP-A) spacecraft. Wave measurements were carried out using the Electric and Magnetic Field Instrument Suite and Integrated Science (EM-FISIS) [22]. Fluxes of particles with energies of a few eV to tens of keV were recorded by the Helium, Oxygen, Proton, and Electron (HOPE) Mass Spectrometer [23]. The fluxes of electrons and protons (as well as helium and oxygen ions) were measured at 11 pitch angles from 4.5° to 175.5° and in 72 energy channels with a time resolution of ~20 s. The energy range is from 15 eV to 50 keV for electrons.

## 3. Overview of the Event 17 March 2015

We analyzed the conjugate measurements between the Van Allen Probe A (VAP-A) and the aurora observations made by the MAIN system in Apatity, Kola Peninsula. The analysis was conducted for the event that occurred on 17 March 2015, from 17:30 to 21:00 UT. Figure 1 displays the trajectory of VAP-A in the magnetosphere, along with the field-aligned projections of the VAP-A satellite onto the ground in the northern hemisphere. The positions of APT and LOZ are indicated on the map by stars. The blue dashed oval represents the boundary at 70° from the zenith in the field of view of the APT all-sky camera.

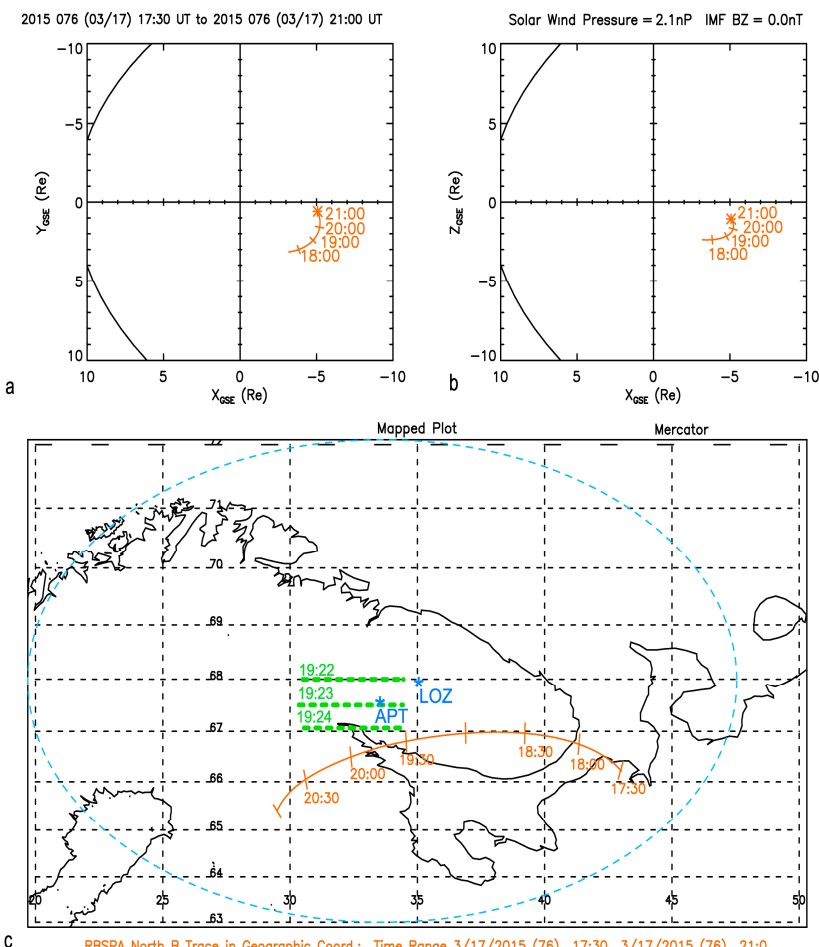

**Figure 1.** Geometry of observations in the case of 17 March 2015: (**a**,**b**) trajectory of VAP-A in the magnetosphere; (**c**) field-aligned projections of the VAP-A satellite in the northern hemisphere in geographic coordinates. The location of APT and LOZ ground stations are marked by blue stars. Blue dashed oval denotes field of view of APT all-sky camera. Location of the discussed auroral forms marked by green dashed lines.

During the event, the VAP-A satellite moved from L = 5.7 to L = 6.6 and then returned to smaller L shells at L = 5.7. The satellite was located in the pre-midnight sector from 20.3 to 22.7 magnetic local time (MLT) at a geomagnetic latitude (MLAT) ranging from −15° to −10° (negative MLAT corresponds to the southern hemisphere).

The geomagnetic activity during this time was fairly high, with the SYM-H index at −150 nT and Kp = 7+. The satellite projection along the magnetic field line, based on the Tsi-89 magnetosphere model, was relatively close in geographic latitudes to APT and LOZ. However, the conformity of patterns in electron flows, which will be discussed below, serves as a better indicator of the projection than a distorted magnetic field under magnetic storm conditions. In such conditions, a few tenths of a degree of latitude provide a good fit for the projection.

The variations of the three components of the magnetic field in LOZ for the event on 17 March 2015 are presented in Figure 2a. It can be observed that two intense substorm activations occurred during the given time period. The first substorm activation started around 17:35 UT, followed by the second one around 19:57 UT. Both activations were accompanied by an increase in auroral light intensity.

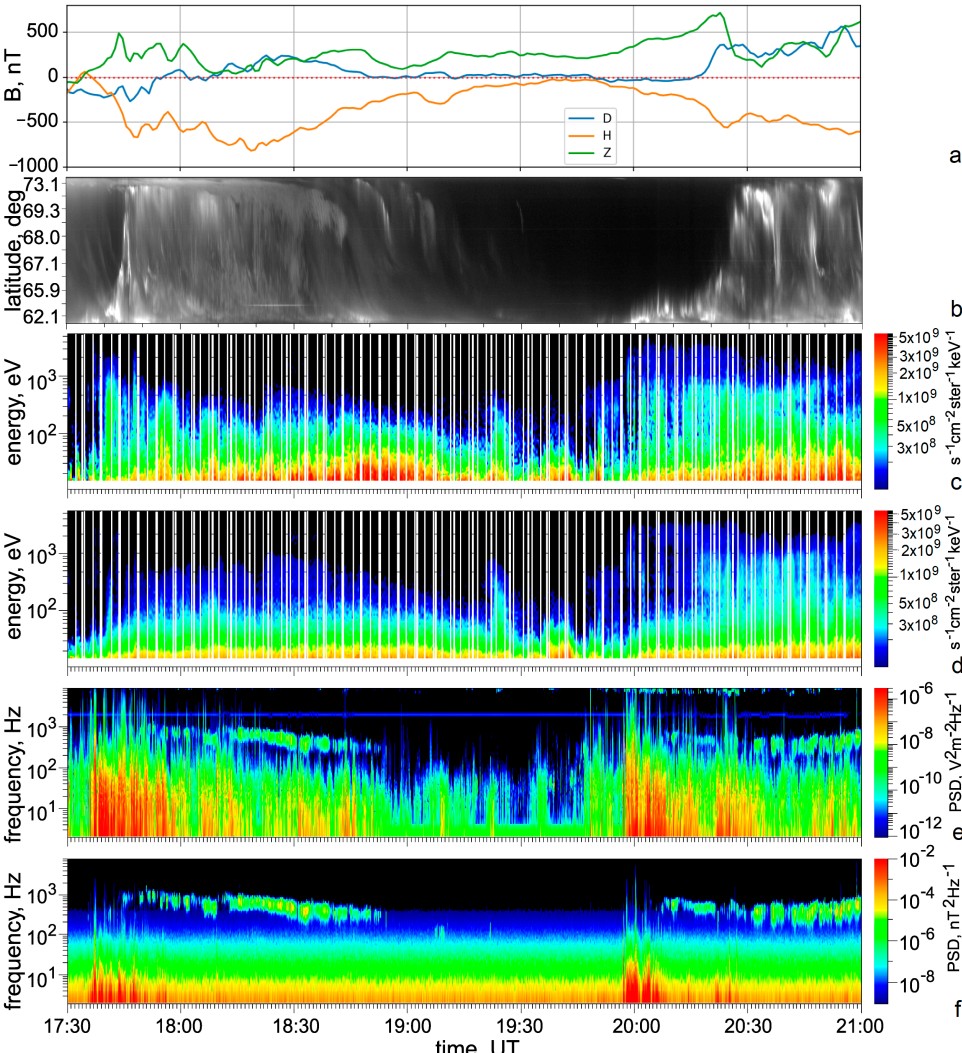

**Figure 2.** Overview of ground-based and VAP-A satellite observations for the event 17 March 2015: (**a**) three components of the magnetic field in Lovozero, (**b**) auroral keogram in the north-south direction of the all-sky camera in Apatity, (**c**,**d**) differential electron fluxes for angles 162° and 90° were obtained on VAP- A satellite, (**e**,**f**) electric and magnetic field spectrograms were detected by VAP-A satellite.

Figure 2b displays an auroral keogram for these events, which illustrates the changes in the glow along the north-south section within the field of view of the all-sky camera in Apatity. At the beginning of the substorm, auroras were observed at lower latitudes, and during the active phase, the area of auroral light expanded and rapidly moved poleward.

As shown in Figure 2b, prior to the onset of the activations, in the region poleward from the intense aurora, several weak auroral arcs slowly moved from north to south in a relatively dark sky during the intervals of 17:30–17:45 and 19:00–19:30 UT.

The arcs depicted on the map in Figure 1 are elongated in the east-west direction and are represented by green dotted lines. The satellite projection moves towards the west with a slight southward offset. The satellite is expected to intersect the magnetospheric structures corresponding to these arcs, and this intersection should be evident in the measured characteristics. Later in the paper we will demonstrate the concomitance of the aurora with precipitating electrons and electrostatic waves observed on the satellite.

During both substorm activations, the VAP-A satellite recorded intense electron fluxes, with the perpendicular fluxes being much smaller compared to the electron fluxes at angles close

to the magnetic field, particularly for low-energy electrons (<100 eV). Please refer to Figure 2c,d, which display the differential electron fluxes for angles of 162° and 90°, respectively.

Simultaneously with the development of the disturbances, electric and magnetic antennas recorded various wave modes. These included whistler waves with frequencies ranging from 100 to 1000 Hz, electron cyclotron harmonic waves with frequencies above 9 kHz, kinetic Alfven waves (characterized by the magnetic component with frequencies below 10 Hz), and broadband electrostatic waves with the highest amplitude at frequencies below 100 Hz.

## 4. Determination of the Auroral Glow Altitudes and the Electron Energy

In the general case, stereometric triangulation does not allow for the reconstruction of the volume distribution of the glow. However, in special cases where additional information about the glow region is available, this becomes possible. Specifically, for auroral rays that are elongated along the magnetic field lines in a column-like shape, it is possible to determine the altitude profile of the volume emission rate along these rays.

The main requirement in this case is the ability to isolate the contrasting structure of the glow in order to identify it in the frames captured by both cameras. By comparing these frames, the altitude at which the structure is located can be determined by observing its displacement against the background of distant stars. A detailed description of this technique can be found in [16]. The altitude profile of the volume glow can then be used to estimate the energy spectrum in the flux of electrons that caused this glow [17,18].

In the interval being considered, our focus is on the rayed auroral arcs that traverse the field of view of the cameras, elongated in the east-west direction, prior to the onset of the active phase of the auroral disturbance. These arcs slowly move from north to south. The intersection of such a structure was observed on the cameras of the MAIN system at approximately 19:23 UT, as depicted in Figure 1, indicated by green dashed lines. Prior to the first activation (before 17:40 UT), these arcs did not exhibit noticeable rays; thus, the triangulation technique had no chance to be applied. Three arcs were observed between 19:10 and 19:30 UT, with rays clearly visible in the arc from 19:20 to 19:27 UT. In [24], similar rays in a comparable case were referred to as "beads". To comprehend the mechanism behind the appearance of these arcs and beads, it is necessary to estimate the energy of the electrons responsible for the glow.

Figure 3 (left column) displays the 4 pairs of frames in negative, rotated to align the horizontal direction with the line connecting the cameras [25]. The frames highlight the rays for which triangulation is performed. The blue crosses indicate the positions of the local magnetic zenith in the frames. The altitude of the points along the ray is determined geometrically by the offset relative to the stars, meaning that the registered glow intensity value is not directly utilized. Consequently, the integrated glow intensity along the line of sight for the right and left frames in each pair is measured independently.

Taking into account corrections for the geometry of the glow region, the angle between the line of sight and the auroral ray, effective radius of rays (~0.5 km [16]) and the effective calibration coefficient for these cameras [20], we obtained volume emission rate profiles [16]. Figure 3 (right column) shows the volume emission rate profiles obtained for the right and left frames. Sharp outliers correspond to the intersections of bright stars. Taking this into account, there is good agreement between the profiles. The obtained profiles have a wide maximum of 200–260 km.

The experimental profiles are used to estimate the energy distribution in the electron flow, which causes such a glow [17,26]. Some uncertainty is the real unknown state (composition) of the atmosphere at high altitudes, which strongly depends on the level of disturbance. For estimates, we took the atmospheric profile from the MSIS model [27].

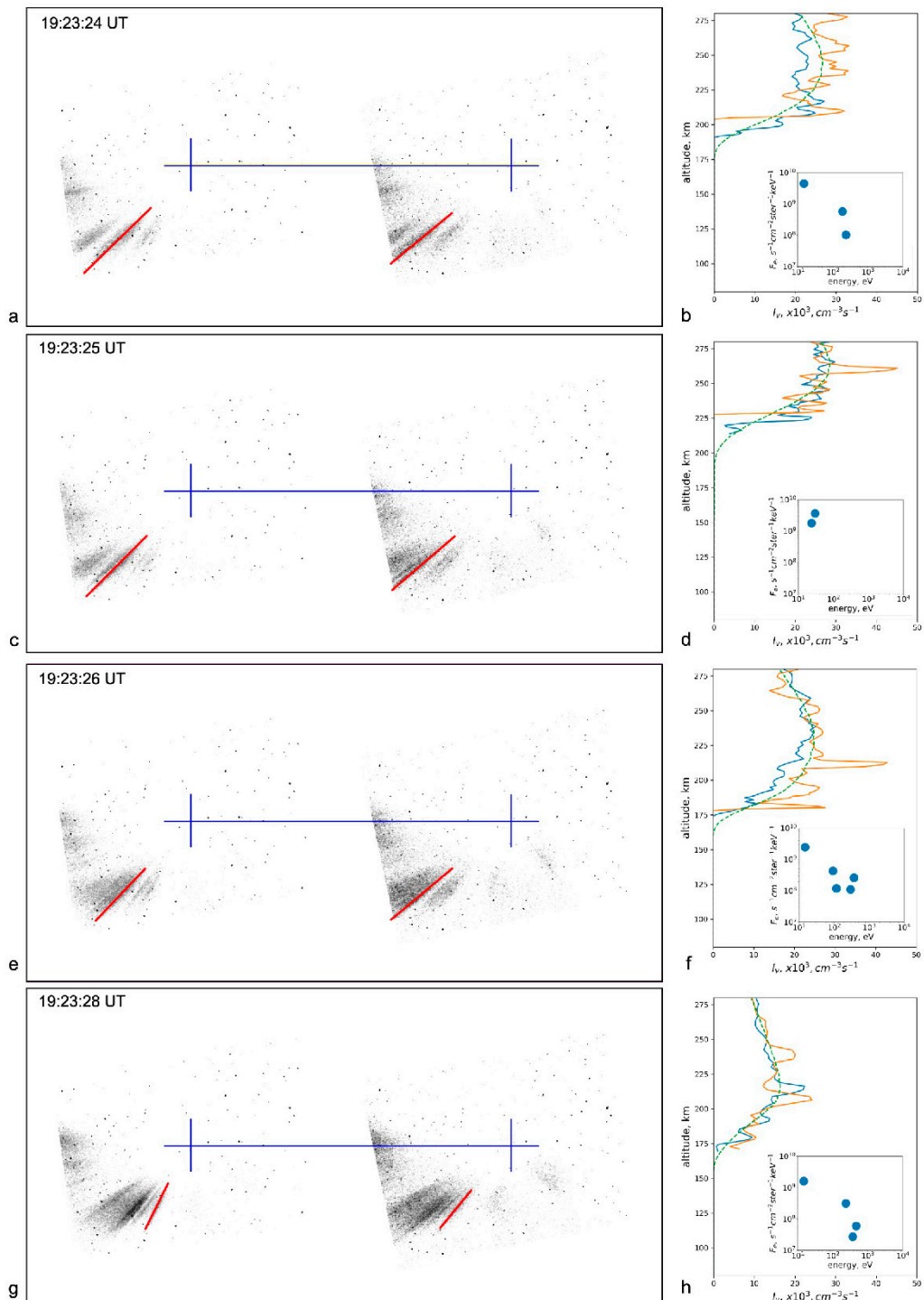

**Figure 3.** Triangulation of the auroral rays: (**a**,**c**,**e**,**g**) pairs of frames from narrow field of view cameras (negative), rays for which triangulation is carried out are highlighted, blue crosses indicate the position of the local magnetic zenith; (**b**,**d**,**f**,**h**) profiles of volume emission rate for right and left frames (solid lines), model profile is a dotted line. Energetic spectra of electrons causing such model profiles shown in the insets.

The energy distribution was estimated by fitting a combination of monoenergetic electron beams in an isotropic pitch-angle distribution to the lower hemisphere, in order to reproduce the average altitude profile for each experimental pair. The problem of electron transport in the atmosphere for auroral electron energies was solved using the method

described in [26,28], which employed analytic approximations of the results obtained from extended Monte-Carlo calculations of the transport of auroral electrons in the atmosphere. The resulting model profiles are shown in Figure 3 (on the right panels) with a dotted line. The fitted energy spectra are shown in the insets. In all cases, the primary contribution to the energy spectrum of electrons comes from a peak with an energy of 50–400 eV, corresponding to the altitude of the maximum in the profile, with an additional contribution at energies of a few tens of eV, extending the profile to higher altitudes.

## 5. Pitch-Angle Distributions of Electrons of Different Energies on VAP-A Associated with Aurora

It is interesting to compare the energies of precipitating electrons determined in the rays with the characteristics of electrons in the equatorial region on the VAP-A satellite. Figure 4 shows, from top to bottom, the energy spectrum of the electron flux at a pitch angle of 162° (upper panel) and the pitch angle distributions of electrons in energy channels of 15, 27, 47, 149, and 260 eV, registered on the satellite in the interval 19:10–19:30 UT.

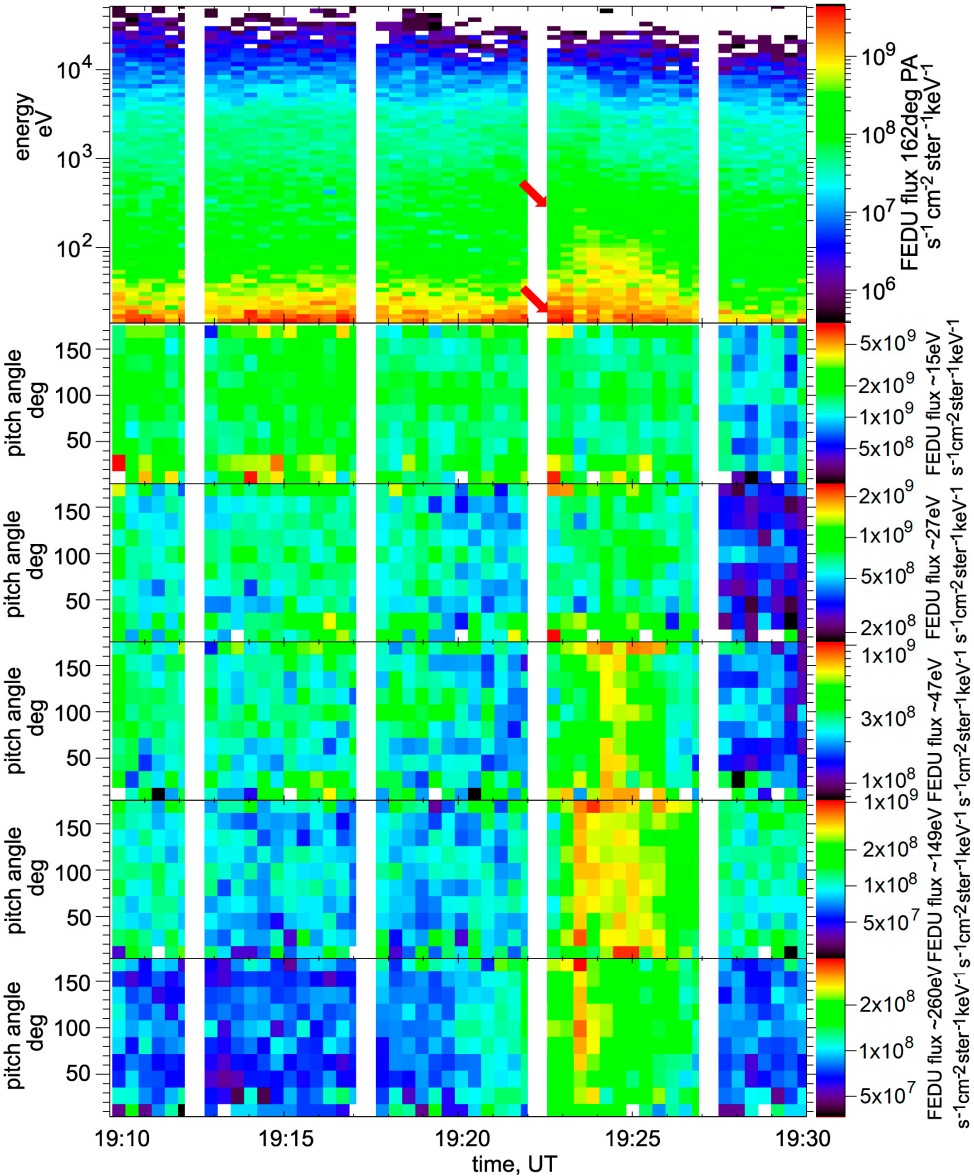

**Figure 4.** The energy spectrum of the electron flux at a pitch angle of 162° (upper panel) and the pitch angle distributions of electrons in energy channels of 15, 27, 47, 149, 260 eV, registered by the VAP-A satellite in the interval 19:10–19:30 UT, 17 March 2015.

Figure 4 shows that the most intense electron flux was recorded from 19:21 to 19:26 UT, which coincides with the time when rays were observed in the aurora. On the top panel of Figure 4, it can be seen that the electron flux near the loss cone (angle channel 162°) was recorded at energies below 400 eV. The maximum flux, $J \sim 4 \times 10^9$ s$^{-1}$cm$^{-2}$ster$^{-1}$keV$^{-1}$, was observed for small electron energies below 30 eV, and the electron flux rapidly decreases with increasing energy. The pitch-angle distributions of electrons in individual channels show that the flux of low-energy electrons with E < 30 eV increases mainly at small angles, but for E > 50 eV, it covers a wider range of pitch angles.

Note that during the increase in electron flux, the energetic spectrum often contains two populations (shown by red arrows): electrons with E < 30 eV, whose fluxes decrease with increasing energy, and more energetic electrons at E ~ 50–400 eV. On the top panel of Figure 4, during the interval of 19:24–19:25 UT, local peaks are visible at energies of 50–100 eV. An analysis of the electron spectra showed that local peaks at E ~ 200–400 eV were also detected in the interval 19:22:30–19:23:30 UT. Due to the lower intensity of electron fluxes at energies >100 eV, these peaks are faintly visible on the border of green and yellow colors on the top panel of Figure 4.

Hence, one can expect precipitation of electrons at low altitudes in the regions conjugated with VAP-A with energies of 10–400 eV, which correspond to the energies of electrons observed in auroras. The presence of local maxima in the spectrum of precipitating electrons in the range of 50–400 eV indicates the presence of two populations in the spectrum of precipitating electrons: electrons below 50 eV and higher-energy electrons.

## 6. Auroral Particle Precipitation and Broadband Electrostatic Waves on VAP-A Satellite

Before the onset of the substorm at 20 UT, three small intensifications of aurora light were observed in the interval 19:08–19:25 UT. Figure 5a presents a detailed auroral keogram, which shows the change in glow in the north-south section within the field of view of the all-sky camera in Apatity during this time interval. The red arrows indicate these light increases.

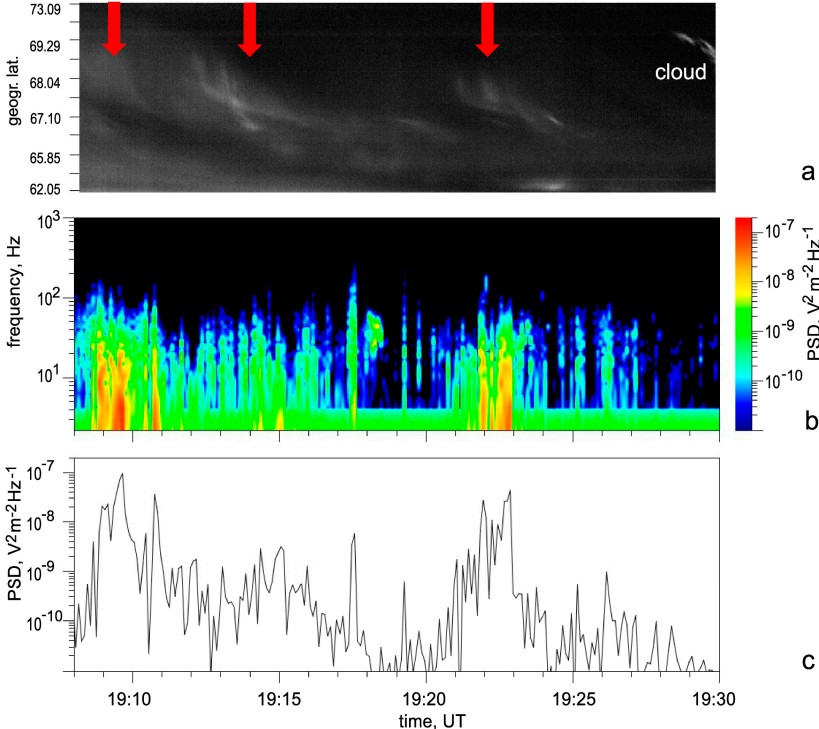

**Figure 5.** Comparison of satellite and ground-based observations on 17 March 2015: (**a**) keogram in the north-south section of the all-sky camera in Apatity; (**b**) spectrogram of the wave electric field observed by VAP-A; (**c**) the amplitude of the electric field at the frequency ~ 4 Hz observed by VAP-A.

Figure 5b,c present the spectrogram of the wave electric field and the amplitude of the electric field at approximately 4 Hz, respectively. A comparison of the panels in Figure 5 shows that in the interval 19:08–19:25 UT, three variations of aurora glow are accompanied by well-isolated broadband electrostatic wave amplifications with a maximum amplitude at frequencies < 100 Hz. It is important to note that during this interval, only electrostatic fluctuations were observed on the satellite, and other wave modes were not registered. Thus, it is natural to assume that auroras and field-aligned distributions of electrons were produced by electrostatic waves.

## 7. Discussion and Conclusions

The presented data from simultaneous ground-based and satellite observations in magnetically conjugated regions in the ionosphere and magnetosphere provide evidence for the relationship between electron fluxes that generate auroral rays and the processes of wave–electron interaction in the magnetosphere at 6 Re. Various types of wave modes are known to be responsible for auroral precipitation. In recent years, special attention has been given to high-intensity nonlinear broadband electrostatic waves, which have been detected in the inner magnetosphere near the equator on the Van Allen Probe satellites, as in the case being considered.

We believe that the interaction between waves and electrons occurs in the equatorial region, where electrons are scattered over pitch angles from a population of trapped particles with a maximum at 90°. As a result, a flow of scattered electrons is generated symmetrically in both hemispheres within the loss cones. At some distance along the magnetic field line from the interaction region, the scattered electron flux becomes anisotropic. Electrons with energies of 150–300 eV, observed around 19:22–19:25 UT with angles of 162°–175°, could be particles scattered by waves as they pass the equator. These symmetrically scattered electrons in both hemispheres contribute to the formation of the auroral glow in the form of rays.

In this particular case, the auroral rays exhibit a distinct structure without blending with neighboring structures, allowing for the determination of the altitude profile of the volume emission rate through triangulation observations. Based on this altitude profile, the energy distribution in the electron flux responsible for the glow at these altitudes was analyzed. The resulting energy distributions consist of two components: a peak with an energy range of 50–400 eV, corresponding to the altitude of the maximum in the profile, and an additional component at energies of a few tens of eV, which extends the profile to higher altitudes. These findings are in good agreement with observations of precipitating electron flux into the loss cone at VAP-A. It is worth noting that this is the first time the energies of precipitating electrons in the rays have been compared with the characteristics of electrons and waves in the equatorial region.

It is of interest to make a qualitative comparison between the aforementioned observations of weak perturbations and the case where the perturbation is stronger. Observations conducted after 20:00 UT on the given day provide an opportunity for such a comparison. Previous analyses [16–18] have examined several rayed structures during this period. The associated spectra of precipitating electrons were fitted using a Maxwellian distribution, with an additional power-law spectrum at low energies. In the more perturbed case, a greater diversity of spectra was observed, primarily at higher energies (500–2000 eV) within the auroral rays. Additionally, broadband electrostatic waves were more intense on VAP-A during this time. It is worth noting that, apart from rayed glows, there were also glows without obvious rays during this period. The distinct boundaries of these structures allow for an estimation of their height, which is approximately 110–120 km. Electron fluxes of 5–10 keV are likely responsible for such glows. Furthermore, it is important to mention that different types of wave modes were recorded during the disturbance after 20 UT. For instance, whistler waves observed on the satellite at frequencies around 1 kHz could potentially induce such electron precipitation. However, determining the contribution of each wave mode to electron scattering in a strongly perturbed case is quite challenging.

Our conclusions from this analysis are as follows:

Using the triangulation of the aurora observations, we determined the precipitating electron energies for rays in the weak aurora during the 17 March 2015 event in the time interval 19:22–19:25 UT. We found that the main contribution to the energy spectrum of electrons comes from a peak with an energy of 50–400 eV, with an additional contribution at energies of a few tens of eV, which extends the profile to higher altitudes.

The largest increase in the electron flux was recorded on the conjugate satellite VAP A at the same time when the rays were observed in the aurora. Comparing the spectra of precipitating electrons in the raised structures of the aurora with measurements of electrons in the equatorial region aboard the VAP-A satellite, in combination with observations of the aurora, showed qualitative agreement. First, during the registration of the aurora, the electron fluxes on the satellite inside and near the loss cone increased by E ~ (10–400) eV, which corresponds to the electron energy range determined in the rays of the aurora. Second, the spectrum of precipitating electrons on a satellite contains two populations: electrons with E < 50 eV, whose fluxes decrease with increasing energy, and more energetic electrons with local peaks at E ~ 50–400 eV.

It is shown that the intensification of the aurora was concomitant with broadband electrostatic waves with a maximum amplitude at frequencies <100 Hz, which were detected by the VAP-A satellite in the conjugate region near the equator. Since only broadband electrostatic fluctuations were observed on the satellite during the coincidence of the aurora and waves, we assume that broadband electrostatic waves cause electron precipitation that produces rayed structures in the aurora.

**Author Contributions:** Conceptualization, B.V.K.; investigation, B.V.K. and E.E.T.; auroral camera data curation, B.V.K.; writing—original draft preparation, B.V.K. and E.E.T. All authors have read and agreed to the published version of the manuscript.

**Funding:** This research was funded by the Russian Science Foundation (grant no. 22-12-20017).

**Data Availability Statement:** Data of auroral observations by MAIN camera system are available from http://aurora.pgia.ru/archive.html (accessed on 1 June 2023). The data used in this study are available from the EMFISIS website at http://emfisis.physics.uiowa.edu/data/index (accessed on 1 June 2023) and from https://cdaweb.gsfc.nasa.gov (accessed on 1 June 2023 ). The authors are grateful to the Van Allen Probe team for free access to the data.

**Acknowledgments:** The authors are grateful to the Van Allen Probe team for free access to the data.

**Conflicts of Interest:** The authors declare no conflict of interest. The funders had no role in the design of the study; in the collection, analyses, or interpretation of data; in the writing of the manuscript; or in the decision to publish the results.

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
