# Peer review of "Conjunction Ground Triangulation of Auroras and Magnetospheric Processes Observed by the Van Allen Probe Satellite near 6 Re"

_universe, doi:10.3390/universe9080353_

Round 1

Reviewer 1 Report

Review of "Conjunction ground triangulation of auroras and magnetospheric processes observed by the Van Allen Probe satellite near 6 Re" by Kozelov, et al.

Comments:

1) The captions for Figures 1, 2, 3, 4 and 5 are truncated.

2) Line 215: I'm not sure velocity profile is the right phrase, I think you mean luminosity profile.

3) Line 252: Please give a little more information about the atmospheric transport modeling, not just stating a reference.

4) How conjugate are the ground observations to the VAP measurements?  Please discuss this point in more detail.

The general usage of English throughout the paper needs improving, and needs to be edited for English usage and grammar.  This needs to be addressed before the paper can be accepted for publication.

Reviewer 2 Report

The article "Conjunction ground triangulation of auroras and magnetospheric processes observed by the Van Allen Probe satellite near 6 Re" by Kozelov&Titova is an interesting case study of a weak rayed auroral event occurred on March 17, 2015 on the occasion of substorm activity. After determination of the auroral glow altitude by triangulation, and  assessment of the electron energy spectrum from the velocity profile of the volume glow, the Authors compare such electron energies to energy- and pitch-angle distributions of VAP-A/HOPE low-energy electrons in the near-equatorial region, tentatively matching HOPE e- features to the spectra of auroral electrons. They also find a strict concomitance between specific glow changes in the auroral keogram and a set of amplications of broadband electrostatic waves measured by VAP-A/EMFISIS in the apparent absence of other possible wave confounders.
The work is detailed and supported by a good reference list. The topic fits the scope of the journal and is of interest for the Community.
Yet, some conclusions - especially those stemming from data shown in paragraph 5 - must be reviewed and strenghtened before any publication.
Precisely:

1) The assessment of the energy distribution in the auroral electron flow is made following reference 17. Reading reference 17, I see that the energy spectrum is specified in the form of a Maxwellian distribution with isotropic pitch-angle distributions in the lower hemisphere, since this matches experimental data from Atmosphere Explorer C satellite. Does working with auroras in the northern hemisphere make any difference?

2) Lines 321-325:
The Authors say: "Figure 4 shows that the most intense electron flux was recorded from 19:21 to 19:26 UT, that is, at the same time when rays were observed in the aurora. At this time, the spectrum contains two populations of electrons (shown by red arrows): electrons with E<30 eV, whose fluxes decrease with increasing energy, and more energetic electrons with local peaks at E~100–300 eV. At higher energies, the spectrum falls rapidly."

Yet, in Fig. 4 (first panel), across the time range 19:21 to 19:26 UT, I can actually see local secondary peaks at E~50-100 eV.

3) Lines 332-338:
The Authors say: "As can be seen from Fig. 4 in the lowest energy channel (15 eV) dominated electrons с pitch angles 4.5°-18°, i.e electrons coming from the southern hemisphere. This is probably the secondary electrons from ionization in the upper southern atmosphere. For energies of 149 and 260 eV, fluxes at large angles (162°) predominate, i.e., moving into the southern hemisphere from the equator region."

At a first look, time-resolved pitch-angle distributions of HOPE e- fluxes do not benefit from selection of single energy channels in terms of particle statistics.
I cannot see an unmistakable dominance of 4.5°-to-18° pitch angles in panel 2 of Fig. 4: a sum of 15-eV and 27-eV channels might help.
Also, 149-eV and 260-eV distributions do not appear dominated by large PAs, but rather spanning larger angle ranges.

4) Line 391:
I would not use the word "correlate", since no long-term statistical investigation is presented here. This is a case study, which nonetheless shows that aurora glow variations are concomitant with well-isolated BB electrostatic wave amplifications, suggesting a possible causal link.

English wording and phrasing are good, but several typos must be corrected here and there.
Precisely:
  - Line 31: "frequently are" --> "are frequently"
  - Line 32: "intensive" --> "intense"
  - Lines 52-53: "and significant" --> ", as well as significant"
  - Line 57: "used a simple implementation of the stereo triangulation approach" --> "a simple implementation of the stereo triangulation approach has been used"
  - Line 61: "and to estimate the" --> "and an estimate of the"
  - Line 71: "are directed to" --> "are oriented towards"
  - Line 77: "lights allows" --> "light allowed"
  - Line 80: "had" --> "have"
  - Line 81: "distributions" --> "distribution"
  - Lines 82-83: "were made by" --> "are from"
  - Line 90: "have analyzed of" --> "have analyzed"
  - Line 93: "trajectory of VAP-A" --> "VAP-A trajectory"
  - Line 95: "denote" --> "denotes"
  - Line 101: "with the SYM-H index was" --> "with the SYM-H index at"
  - Line 102: "according" --> "according to the"
  - Line 139 (Fig.1 caption): "denoted" --> "denotes"
  - Line 142: "three components" --> "the three components"
  - Line 148: "At the beginning of both substorm aurora were recorded" --> "At the beginning of both substorms, auroras were recorded"
  - Line 161: "which have" --> "with"
  - Line 218: "in east-west direction" --> "in the east-west direction"
  - Line 223: "cannot be applied" --> "had no chance to be applied"
  - Line 226: "beams" --> "beads" maybe?
  - Line 228: "On Fig. 3 on the left shows" --> "Fig. 3 (left column) shows"
  - Lines 233-234: "the glow intensities integrated intensity" --> "the integrated glow intensity" maybe?
  - Lines 237-238: ", effective radius of rays (~0.5 km [16]) and taking into account the" --> ", the effective radius of rays (~0.5 km [16]), and the"
  - Line 239: "In the right column in Figure 3 shows" --> "Fig. 3 (right column) shows"
  - Line 311 (Fig.3 caption): "model profiles is" --> "model profile is"
  - Line 312 (Fig.3 caption): "causing to such model profiles" --> "causing such model profiles"
  - Line 333: "dominated electrons с pitch angles 4.5°-18°" --> "distributions are dominated by electrons with 4.5°-to-18° pitch angles"
  - Line 334: "This is" --> "These are"
  - Line 384: "Fig. 6a" --> "Fig. 5a"
  - Line 388: "spectragrams" --> "spectrogram"
  - Line 460: "The spectra of precipitating electrons for them" --> "The associated spectra of precipitating electrons"
  - Line 464: "At this time broadband electrostatic waves was also" --> "At that time broadband electrostatic waves were also"
  - Line 465: "this period" --> "that period"

Round 2

Reviewer 1 Report

The authors have adequately addressed by comments and the usage of English was dramatically improved in this version.

Author Response

The authors would like to thank the reviewer for the positive evaluation of the made corrections and the article as a whole.

Reviewer 2 Report

I'm pretty fine with Kozelov&Titova's rebuttal - I can recommend publication.
It would be interesting if any study continuation can be made, in order to pass from case study to long-term investigation and possibly outline a general behavior.
One last warning concerns figure captions, which appear somewhat truncated across the entire text - it should be just a matter of messy TeX compilation.

Author Response

The authors would like to thank the reviewer for the positive evaluation of the made corrections and the article as a whole. Such research requires successful spatial alignment of ground observation geometry, satellite passes, and geomagnetic disturbance levels. Of course, we will continue to search for interesting cases that can be analyzed. The issue of caption formatting will be resolved with the editor, as it appears differently in different programs.